# Perception of the Lebanese Adults about Vaccination: A Survey

**DOI:** 10.3390/vaccines11030621

**Published:** 2023-03-09

**Authors:** Rania Sakr, Mariana Helou, Cima Hamieh, Michelle Estephan, Remie Chrabieh, Pascale Salameh, Georges Khazen, Harout Kolanjian, Elsy Jabbour, Rola Husni

**Affiliations:** 1Lebanese American University Medical Center Rizk Hospital, Division of Family Medicine, Lebanese American University School of Medicine, Beirut 1102 2801, Lebanon; 2Lebanese American University Medical Center Rizk Hospital, Division of Emergency Medicine, Lebanese American University School of Medicine, Beirut 1102 2801, Lebanon; 3Department of Dermatology, Lebanese American University Medical Center Rizk Hospital, Beirut 1102 2801, Lebanon; 4Faculty of Medicine, Lebanese American University, Beirut 1102 2801, Lebanon; 5Computer Science and Mathematics Department, Lebanese American University, Beirut 1102 2801, Lebanon; 6Lebanese American University Medical Center Rizk Hospital, Division of Infectious Diseases, Lebanese American University School of Medicine, Beirut 1102 2801, Lebanon

**Keywords:** adult vaccination, vaccine, immunization, global health

## Abstract

Introduction: With the continuous spread and emergence of transmissible diseases, focusing on preventive measures is essential to decrease their incidence and spread. In addition to behavioral measures, vaccination is an optimal way to protect the population and eradicate infectious diseases. The majority are aware of children’s vaccinations, while many might not know that adult vaccinations are also essential. Objectives: This study aims to understand the perception of Lebanese adults towards vaccination and their knowledge and awareness of its importance. This is a national cross-sectional study, conducted between January 2020 and January 2021. Results: the data were collected from 1023 subjects, the majority being Lebanese, previously healthy, and with a graduate or post-graduate level of education. Out of these participants, 44.9% were advised to take vaccines, half of them by healthcare workers. The most common vaccine received during adult life is the Flu vaccine. Overall, 25.6% of the participants were unaware that they needed vaccines and 27.9% thought it is not indicated. Participants’ knowledge about vaccination is variable. In total, 39.4% agree or are uncertain whether vaccines contain harmful chemicals and 48.4% believe that vaccines will trigger diseases. The level of education and occupation significantly enhances knowledge about vaccination. Some participants 27.3% are concerned about the vaccine’s side effects. The group of young participants, graduates, and nonsmokers think that the vaccine is a necessity and had a positive attitude towards vaccination. Conclusions: Many Lebanese lack knowledge about adult vaccination protection and its benefits in the community. It is essential that the country’s health ministry department collaborate with the healthcare system to launch awareness campaigns about adult vaccination in the country to overcome the barriers and ensure better coverage.

## 1. Introduction

Vaccines have globally been successful in limiting transmissible diseases. In fact, immunization has become the most effective strategy to prevent these diseases [1]. The efficacy of vaccination is not only reflected by the significant decline and even eradication of many transmissible diseases, but also by the decrease in the incidence of related morbidities [2,3]. Vaccination has globally become a component of routine care for children, however, this is not the case for adults [4]. Although the adult schedule for vaccination is updated on a yearly basis by the Advisory Committee on Immunization Practices ACIP [5], vaccine coverage remains low among adults for most of the recommended vaccines [6,7]. Several vaccines given during childhood require a booster dose in adulthood, while others are targeted for adults. Vaccination programs should become mandatory for adults as they are for children, and should be as important as a preventative measure [8].

It is then essential to assess, people’s knowledge and attitudes towards adult vaccination to identify the possible barriers. Many of those barriers have been documented in the literature including factors related to the vaccine itself, the targeted disease, the social influences, the awareness, and the knowledge or cultural habits [6,7,8,9].

In one study conducted in the United States in 2015, adults completed an online survey showing that 25% to 72% of them received specific vaccines. Awareness about vaccine-preventable diseases, varied between 63% to 94%, while awareness about their corresponding vaccines varies between 60% to 94%. Awareness level was significantly higher in females, healthcare professionals, and college graduates [10]. In another study conducted in Pakistan in 2014 to assess awareness toward vaccination, the majority of the participants agreed on the safety and necessity of vaccination in adults [8], and reasons reported for refusal of vaccination were: cost at 33%, allergies at 11%, pain or discomfort at 10.8%, and fear of vaccination unsafety at 4%. A Canadian study conducted in 2010 showed similar results among 4023 adults where 57% of participants and 88% of healthcare professionals agreed on the importance of vaccination, reflected by the fact that 46% of adults and 75% of healthcare providers were up-to-date on their vaccines [4]. To note, several studies have documented that better knowledge about vaccination improves the vaccination rate [11,12].

Lebanon is a small country in the Middle East. The healthcare system is mainly a private system. Lebanon’s economy has been collapsing since 2019 and its currency has lost more than 90% of its value. There is no published data about adult vaccination in the country. However, after the COVID-19 pandemic, data was collected. The latest data of 17 February 2023, reveals the vaccination rate for three doses of COVID-19 to be 27.4% [13]. No studies about vaccination were conducted in Lebanon. Therefore, this study was conducted to assess adult vaccination awareness and barriers, in order to highlight all gaps to be able to achieve better coverage.

## 2. Methods

### 2.1. Sample and Data

This is a national cross-sectional study, conducted between January 2020 and January 2021. This study was approved by the Lebanese American University Institutional Review Board IRB # LAUMCRH.RH2.7/Dec/2018. Adults visitors and patients presenting to the clinics or the Emergency of the Lebanese American University Medical Center-Rizk Hospital (LAUMCRH) between January 2020 and January 2021 were asked if they are willing to participate and if agreed, were included in the study. Family medicine and Emergency Residents were approaching the participants and assisted them in filling out the questionnaire (Appendix A). English copies of the questionnaire were electronically circulated. Written informed consent has been obtained from the subjects to publish this paper.

### 2.2. Measures of Variables

The questionnaire was composed of two parts: the first part was demographic data which included age, level of education, occupation, being a healthcare worker or not, area of residence, and family status. The second part were questions about the knowledge and attitude toward vaccination. Questions were derived from previously available online questionnaires studying vaccination knowledge in different populations [2,4,6]. The survey was validated and piloted by testing it on random subjects. Taking an acceptable risk of error of 3%, a 95% confidence level, and a total population of 5 million, the minimal sample size was calculated to be n = 998 on Epiinfo software. We successfully managed to obtain the response of 1023 individuals which meets the sample size requirements of this study.

### 2.3. Data Analysis Procedure

The results of the survey questionnaires are presented as raw counts and crude frequencies. Age groups were divided into three categories: <30, 30–50, and >50 years old. Relationships between the demographic categorical variables and knowledge, attitude, and practice, were analyzed using the Chi-squared test and a *p*-value < 0.05 indicated a significant association. All statistical analysis was conducted in R (version 4.0.2)

## 3. Results

Data was collected from 1023 subjects. The sample size achieved was calculated to be representative of the population. The majority of these participants were Lebanese (n = 985, 96.3%), single (n = 674, 65.9%), and previously healthy (n = 879, 85.9%). The number of smokers (n = 651) exceeded the number of non-smokers (n = 372). The study included slightly more females than males (Table 1).

Concerning the level of education, the majority of the participants were at the graduate 45.9%, or postgraduate level 24.8%. The majority of participants were healthcare professionals 27.6%, followed by business professions 18.1%, architecture 11.2%, and the remaining professions at a lower frequency.

Our questionnaire included questions that reflect the society’s knowledge about vaccination. Out of the 1023 participants, 459 (44.9%) were advised to take vaccines, of whom 233 (50.76%) were advised by healthcare workers (Pediatricians, Family physicians, Obstetric gynecologists, or Pharmacists). The remaining participants knew about vaccinations needed through awareness campaigns and work requirements.

The most common vaccines recommended were the Flu vaccine, and the Human Papillomavirus.

The most common vaccine received during adult life is the Flu vaccine (39.7%) alone, or with other vaccines. Overall, 25,6% of the participants were unaware that they needed vaccines and 27.9% thought it is not indicated. The majority of the participants (71.8%) are not required to do adult vaccines by their employer, even though 74% agree that adult vaccination is necessary.

Participants’ knowledge of the advantages and side effects of vaccination is variable. In total, 403 participants (39.4%) agree or are uncertain whether vaccines contain harmful chemicals and 496 participants (48.4%) either believe that vaccines will trigger diseases such as diabetes, and autism or are uncertain about this fact (Table 2).

Age, the area of residence, being a parent, and previous medical history were at no point significantly associated with the knowledge. However, the level of education and occupation significantly enhances knowledge about vaccination being safe and protective (Table 3). Our results showed that educated subjects mainly graduates (n = 343, 33,49%) know more about vaccine-preventable diseases and their safety (n = 211, 20.6% for question two and n = 266, 25.97% for question three). As for occupation, with the exclusion of healthcare workers, architects (n = 317, 30.95%) and subjects in the business field (n = 129, 12.59%) has the best knowledge about the protective role of vaccines. Nonsmokers (n = 355, 34.66%) were more aware of the side effects of the vaccines. Participants from the five main Lebanese districts did not differ in their knowledge about vaccination.

Participants’ attitude toward vaccination was also evaluated. In our study, 279 participants (27.3%) are concerned about the vaccine’s side effects and 223 (21.8%) are uncertain about it. Similarly, 227 (22.2%) think that vaccines are not important in adulthood as in childhood and 248 (24.2%) are uncertain about this fact. The majority (78.8%) accepted to receive any of the mentioned vaccines if advised about it and 76.1% would recommend it to others (Table 4).

Being previously healthy had no impact on population behavior in receiving or recommending vaccines. In people less than 30 years old, 51.85% (n = 531) disagree with question one and think the vaccine is a necessity, the same for graduates (n = 356, 34.76%), architects in occupation (n = 336, 32.8%) and nonsmokers (n = 536, 52.34%), hence these groups had a positive attitude towards vaccination, as in Table 5. Parents with no child were more aware of the vaccine’s importance in adulthood (n = 426, 41% vs. n = 122, 11.9%).

To decrease bias, healthcare professions were not included in the analysis of this section. Architects and entrepreneurs had the most positive attitude towards vaccines’ importance and safety. As for the side effects, graduate-level students (n = 204, 19.9%) are the least concerned about the safety of the vaccines. The people living in Beirut and Mont-Lebanon were the ones who showed the least concern for the side effects, (n = 241, 23.53% and n = 185, 18.06% respectively).

As to who actually received their vaccinations, 492 participants (48.04%) took at least one of the adult vaccines (Td, Flu, pneumococcal), the most common being the yearly influenza vaccine. Surprisingly, only 403 participants (39.35%) received the Flu vaccine. A total of 532 participants (51.9%) were not updated on their vaccines schedule. The reasons were unawareness (n = 262 or 44.94%), thinking it is not indicated (n = 285 or 48.8%), fear of the side effects (n = 21 or 3.6%), and financial concern (n = 14 or 2.4%).

Finally, when asked about their preferred way of receiving information about vaccination, most (74%) selected social media platforms, 8% liked their doctor to tell them about vaccination, 6% picked billboards, and 5% favored brochures and Television.

## 4. Discussion

In our sample of 1023 participants, the majority were Lebanese, previously healthy, and of graduate or post-graduate level of education. When asked, 45% of the participants were advised to take vaccines, half of them by their healthcare workers. The most common vaccine received during adult life is the Flu vaccine alone or combined with others. Overall, 25.6% of the participants were unaware that they needed vaccines and 27.9% thought it is not indicated. Participants’ knowledge about vaccination is variable. In total, 39.4% agree or are uncertain whether vaccines contain harmful chemicals and 48.4% believe that vaccines will trigger diseases. The level of education and occupation significantly enhances knowledge about vaccination. Some participants, 27.3%, were concerned about the vaccine’s side effects. The group of young participants, graduates, and nonsmokers think that the vaccine is a necessity and had a positive attitude towards vaccination.

While most people in developed countries have a positive approach towards adult vaccination [14,15], the situation in Lebanon does not seem to be the same. It is clear from our study that the great majority of subjects, who never received a vaccine as adults, were not aware of its importance or believed it was not indicated. These findings are similar to the ones discussed in the Canadian study indicating the need for educational intervention [14]. These results would have certainly changed if the study was conducted years later, after the COVID-19 vaccine, knowing that the COVID-19 vaccines have unluckily increased the hesitancy towards immunization practice. A survey conducted on 1012 adults in the United States before the COVID-19 vaccine was available showed that 68% agreed to get the vaccine for themselves and 65% would vaccinate the adults under their care [16]. However, in a study carried out among refugees and Lebanese nationals pre and post-COVID-19 vaccination [17], the vaccine acceptance was very low in both groups in the pre-vaccination survey (around 25%); and the acceptance rate was higher after the vaccination [17]. The Lebanese nationals had significantly higher vaccine acceptance compared to the refugees [17].

The results of this survey that might be of concern are the lack of knowledge about vaccines. However, around half of the participants admitted to being advised, mostly by health care professionals, to take a vaccine at least once in their adult life, many lacked knowledge about adult vaccination and its importance. This reflects the important role of physicians and other health care providers play in the awareness about this topic.

In fact, around 40% agree that vaccines contain harmful chemicals, and around 50% either believe that vaccines will trigger diseases such as diabetes or autism, or are uncertain about this fact. In further analyzing these findings, we realized that the majority of people who had better knowledge about vaccination were non-smokers, probably this knowledge affected positively this behavior because of their knowledge or fear of illness related to smoking.

Another factor that affected awareness is the educational level. University undergraduates and graduates were more knowledgeable probably because of their easier access to scientific sites and campaigns and because of the possible integration of this information into their curriculum. Of note, at some universities, a screening medical visit is mandatory for all students where a vaccination update is performed. Unexpectedly, postgraduates showed a lower level of knowledge emphasizing the need for awareness campaigns targeting this category. Similarly, for the COVID-19 vaccine, having a higher education is associated with better vaccine awareness [18].

Vaccine hesitancy is variable among countries and depends on many factors, it ranged from 9.3% to 43.2% in one study [19]. The rate of COVID-19 vaccinated people increases with the wealth of the different nations, but the maximum level of vaccination rate seen was 70% between countries. The remaining 30% is due to the natural hesitancy of people to vaccination [20]. A systematic review was conducted in 2021 on COVID-19 vaccine hesitancy worldwide and included 31 published papers from a total of 33 countries [21]. The highest rates of vaccine acceptance (above 90%) were found in Ecuador, Malaysia, Indonesia, and China, while the lowest vaccine acceptance rates (<30%) were found in Kuwait and Jordan [21]. Male gender and age were significant factors for vaccine acceptance rates in most of the studies [21]. In a study conducted in Israel, men were significantly more willing to receive the vaccine and this willingness increased with age and income and decreased with their level of religiousness [22]. Similarly in Israel, after the COVID-19 vaccination started, almost 65% of the adults were willing to receive the vaccine immediately while the others preferred to wait 3 months and even 1 year (17% and 18%, respectively) [23]. A survey conducted on adults residing in France, and randomly selected in July 2020, showed that vaccine refusal and hesitancy were significantly associated with females, older age, lower educational level, poor compliance with recommended vaccinations in the past, and no chronic conditions [24]. A study conducted in Cape Town discussed the contributing factors to low vaccination and hesitancy, the most common were religious beliefs, misinformation through the internet, waiting for natural immunity to develop, concern over causing pain, and concern for side effects [25].

People living in the capital were more knowledgeable than in other regions, this could be linked to better education rates in the capital as compared to the suburbs, as shown in a Lebanese study published in 2013 [26]. However, the difference was not significant to conclude that the area of residency is a factor that affects Lebanese knowledge regarding adult vaccination.

Looking at the two variables, attitude, and practice, we found that around 30% are concerned about the vaccine’s side effects and around 20% are uncertain about it. Similarly, around 20% think that vaccines are not important in adulthood as in childhood and are uncertain about this fact. The majority (almost 80%) accepted to receive any of the mentioned vaccines, if advised about it and would recommend it to others. Therefore, we can deduce that the lack of knowledge pointed out earlier, does not mirror the attitude of the people. People tend to have a positive reaction towards immunization if they are advised about it. We also found that similarly the younger age group, less than 30 years old, had a positive attitude towards taking the vaccines regarding their importance and safety. This could be due to the short interval of time elapsed since their childhood and adolescence. In addition, the awareness offered during their university studies and the parents’ influence during their childhood could have played a role.

For people living in Beirut, the statistics showed that they had the best attitude towards vaccination, this could be explained by the availability of vaccines in the capital. As for smokers, their fear of contracting any infection because of their vulnerable lungs and their more frequent medical visits could have driven their positive attitude toward vaccination. Furthermore, aside from healthcare providers, engineers and business specialists were more receptive to vaccination. This finding could be related to the need for constant travel and the insurance companies mandating vaccination, as compared to people in other fields.

Lebanese adults still do not get vaccinated enough against the Flu. Only 39.35% or less than half of the adults in the sample received their flu shot in the studied year, which is actually comparable to the rate in other countries. In the US, for the same age group, 48% of the people were vaccinated and in Europe, 45% of young adults and 70% of the elderly received the Influenza vaccine [27].

A concern for the population is the protection offered by vaccines as well as the need for booster doses. Even after knowing that vaccination provides a high level of protection, people have fear of the vaccines and are questioning their efficacy especially when we deal with new vaccines, as with the COVID-19 vaccination [28,29,30]. The reluctance of people to receive the recommended vaccines was already a growing concern before the COVID-19 pandemic [31]. However, after the COVID-19 pandemic, this issue became more common. Many studies documented high rates of hesitancy regarding the COVID-19 vaccination among subjects [18,23,25,28,29,32,33]. This phenomenon of COVID-19 vaccine hesitancy appeared more severe in the Middle East and North Africa region, Europe and Central Asia, and Western/Central Africa [32]. Concerns about vaccine safety and side effects were the most common reason for hesitancy [18,29,31,34]. Other causes of concern were the importance and effectiveness of the vaccine [18,34]. Despite the actual financial crisis in Lebanon, financial barriers to the vaccine were not described as concerns [34]. To face this, programs for public confidence in vaccination have been launched and their effectiveness was studied [35].

Since the COVID-19 pandemic continues to be a major threat to society with new mutations, a vast vaccination campaign for the population is the only solution to cope with this [36,37]. Strategies to prevent pandemics in general, and COVID-19 in specific rely mainly on multi-level governance [38,39]. Good governance improves the effectiveness of vaccinations and reduces fatality rates [37,40]. Findings suggested that prevention strategies for COVID-19 infection should be planned and prepared in the summer season to be implemented in the autumn and winter seasons [41].

## 5. Conclusions

A limitation of our study is the fact that it was conducted in one center, however, the samples were diverse enough to provide a good representation of the different Lebanese districts. Another limitation of the study is the vague answer options given in the questionnaire, such as the exact meaning of “completed vaccination” and the definition of unawareness: vaccine existence and whether it is indicated. However, these were explained by the team who was collecting the data.

An important limitation is the paucity of metabolic covariates, for example, obesity, diabetes, and other cardiovascular risk factors that may influence vaccination efficacy [42].

It is essential that the ministry of health and the healthcare system collaborate in launching awareness campaigns about adult vaccination in the country to overcome the barriers and ensure better coverage. Social media platforms would be the most effective way to reach out to the largest number of the population. Strategic educational campaigns, in collaboration with the ministry of health and the World Health Organization, will help increase the rate of vaccination, especially during a pandemic such as COVID-19 where this could be the only way out. This is why good governance and appropriate communication with effective vaccination plans is the only way to avoid and control future pandemics.

## Figures and Tables

**Table 1 vaccines-11-00621-t001:** Descriptive Demographic data (N = 1023), LAUMCRH, January 2020–January 2021.

Gender Male Female	Number (Percentage) 445 (43.5) 578 (56.5)
Marital Status Single Married Divorced Widowed	674 (65.9) 328 (32.1) 13 (1.3) 8 (0.8)
Nationality Lebanese Other	985 (96.3) 38 (3.7)
Previous Medical history Healthy 1 RF 2 RF 3 RF	879 (85.9) 123 (12) 17 (1.7) 4 (0.4)
Smoking Smokers Non-Smokers	651 (63.6) 372 (36.4)
Parents Yes No	267 (26.1) 756 (73.9)

RF: Risk Factor (asthma, lung disease, diabetes, kidney disease, coronary artery disease, heart failure, steroid treatment, and alcohol use).

**Table 2 vaccines-11-00621-t002:** Knowledge of subjects toward vaccination. LAUMCRH. January 2020–January 2021.

Knowledge Questions	Agree N (%)	Disagree N (%)	Uncertain N (%)
1-Vaccines are highly protective against the diseases they are targeting	806 (78.8)	55 (5.4)	162 (15.8)
2-Vaccines may trigger some diseases such as diabetes, autism, or others as side effects	113 (11)	527 (51.5)	383 (37.4)
3-Vaccines contain chemicals that are harmful to humans	81 (7.9)	620 (60)	322 (31.5)

N: Number (P: Percentage).

**Table 3 vaccines-11-00621-t003:** Knowledge of subjects based on Chi^2^
*p*-value. LAUMCRH. January 2020–January 2021.

Knowledge	Age	Education Level	Area of Residency	Having Children	Occupation	Smoking	Medical History
1-Vaccines are highly protective against the diseases they are targeting	6.05	23.8 ***	8.12	0.42	50.43 *	2.79	7.9
2-Vaccines may trigger some diseases such as diabetes, autism, or others as side effects	1.1	19.19 **	10.65	0.41	45.98 *	10.04 **	2.77
3-Vaccines contain chemicals that are harmful to humans	8.28	14 *	6.53	3.81	47.14 *	2.32	4.5

*** = *p*-value < 0.001, ** = *p*-value < 0.01, * = *p*-value < 0.05.

**Table 4 vaccines-11-00621-t004:** The attitude of subjects toward vaccination. LAUMCRH. January 2020–January 2021.

Attitude Questions	Agree N (%)	Disagree N (%)	Uncertain N (%)
1-Vaccines are unnecessary because we can treat the disease once it occurs	63 (6.2)	815 (79.7)	145 (14.2)
2-I am concerned about the side effects of vaccines	279 (27.3)	521 (50.9)	223 (21.8)
3-If there were any indicated vaccines, I would get them done	806 (78.8)	39 (3.8)	178 (17.4)
4-If there were any indicated vaccines, I would recommend them to others	779 (76.1)	65 (6.4)	179 (17.5)
5-Vaccines are of less importance in adulthood	227 (22.2)	548 (53.6)	248 (24.2)

N: Number (P: Percentage).

**Table 5 vaccines-11-00621-t005:** The attitude of subjects based on Chi^2^
*p*-value. LAUMCRH. January 2020–January 2021.

Attitude	Age	Level of Education	Area Residency	Having Children	Occupation	Smoking	Medical History
1-Vaccines are unnecessary because we can treat the disease once it occurs	23.47 ***	16.96 **	12.23	3.82	63 ***	9.3 **	6.16
2-I am concerned about the side effects of vaccines	2.81	23.13 ***	20.46 **	0.44	56.02 **	3.2	4.94
3-If there were any indicated vaccines, I would get them done	3.58	6.3	3.08	2.86	29.7	4.06	7.14
4-If there were any indicated vaccines, I would recommend to others	6.87	8.57	12.75	2.86	39.96	1.59	2.23
5-Vaccines are of less importance in adulthood	20.53 ***	25.92 ***	12.90	9.6 **	60.16 ***	11.76 **	6.08

*** = *p*-value < 0.001, ** = *p*-value <0.01.

## Data Availability

The data presented in this study are available on request from the corresponding author. The data are not publicly available due to ethical reasons.

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
