# Peer review of "Perception of the Lebanese Adults about Vaccination: A Survey"

_vaccines, 2023, doi:10.3390/vaccines11030621_

Round 1

Reviewer 1 Report

Perception of the Lebanese Adults about vaccination: A Survey

The topics of this paper are  interesting, though well known for a lot of countries. The structure and content must be revised, and results have to be  better explained by authors before to be reconsidered for publication.

Abstract has to clarify empirical results and health and social implications of vaccination policy for the Lebanon. 

Introduction has to better clarify the research questions of this study and provide more theoretical background. Authors have to better describe the role of vaccination to cope with COVID-19, optimal level  of vaccination and factors associated with vaccination. After that they can focus on the topics of this study to provide a correct analysis for fruitful discussion (See suggested readings that must be all read and used in the text). 

Methods of this study is not clear. The section of Materials and methods must be re-structured with following three sections only and same order:

Sample and data

Measures of variables

Data analysis procedure. More details to provide.

Results.

Authors have to discuss if the sample is or not representative of population.

Table 1 has also to indicate percentage of value. It would be better to represent these results with a bar graphs.  

Title of tables has to be put above the table and not below. Title has to provide more information about analyses done and indicate location and period of analysis, always. This aspect for all tables. 

Table 2 and 3. Significance of X2 p-value has to be indicated with *, such as ***=p-value <.001, **=p-value <0.01, *=p-value <0.05

 Discussion. 

First, authors have to synthesize the main results in a simple table to be clear for readers and then show what this study adds compared to other studies. 

Section of Limitations and conclusion can be merged. 

Conclusion has not to be a summary, but authors have to focus on manifold limitations of this study and provide suggestions of health, crisis management  and social policy, as well as how nations can prevent and manage with a good governance and appropriate communication next pandemics with effective vaccination plans and nonpharmaceutical measures of control. 

Overall, then, the paper is interesting, but theoretical framework is weak, and some results create confusion… structure of the paper has to be improved; study design, discussion and presentation of results have to be clarified using suggested comments.

I strongly suggest improving the paper,  by using all comments (suggested papers included to read and use) that I will in-depth verify,  and maybe it can be considered. 

Suggested readings of relevant papers that have to be read and all inserted in the text and references.

Solís Arce, J.S., Warren, S.S., Meriggi, N.F., (...), Mobarak, A.M., Omer, S.B. 2021. COVID-19 vaccine acceptance and hesitancy in low- and middle-income countries, Nature Medicine 27(8), pp. 1385-1394

de Figueiredo, A., Simas, C., Larson, H.J. 2023COVID-19 vaccine acceptance and its socio-demographic and emotional determinants: A multi-country cross-sectional study, Vaccine 41(2), pp. 354-364

Coccia M. 2022. Preparedness of countries to face COVID-19 pandemic crisis: Strategic positioning and underlying structural factors to support strategies of prevention of pandemic threats, Environmental Research, Volume 203, n. 111678,  https://doi.org/10.1016/j.envres.2021.111678.

Sallam, M., Al-Sanafi, M., Sallam, M. 2022A Global Map of COVID-19 Vaccine Acceptance Rates per Country: An Updated Concise Narrative Review. Journal of Multidisciplinary Healthcare15, pp. 21-45

Abou-Arraj, N.E., Maddah, D., Buhamdan, V., (...), Alami, N.H., Geldsetzer, P. 2022Perceptions of, and Obstacles to, SARS-CoV-2 Vaccination among Adults in Lebanon: Cross-sectional Online Survey.JMIR Formative Research, 6(12),e36827

Coccia M. 2022. Optimal levels of vaccination to reduce COVID-19 infected individuals and deaths: A global analysis. Environmental Research, vol. 204, Article number 112314, https://doi.org/10.1016/j.envres.2021.112314

Teitler-Regev, S., Hon-Snir, S. 2022. COVID-19 Vaccine Hesitancy in Israel Immediately Before the Vaccine Operation. Yale Journal of Biology and Medicine95(2), pp. 199-205

Benati I., Coccia M. 2022. Global analysis of timely COVID-19 vaccinations: Improving governance to reinforce response policies for pandemic crises. International Journal of Health Governance. https://doi.org/10.1108/IJHG-07-2021-0072

Shmueli, L. 2022. The Role of Incentives in Deciding to Receive the Available COVID-19 Vaccine in Israel, Vaccines10(1),77

Harrison, E.A., Wu, J.W., 2020. Vaccine confidence in the time of COVID-19. Eur. J. Epidemiol. 35 (4), 325–330. https://doi.org/10.1007/s10654-020-00634-3

Coccia M. 2022. COVID-19 Vaccination is not a Sufficient Public Policy to face Crisis Management of next Pandemic Threats. Public Organization Review, https://doi.org/10.1007/s11115-022-00661-6

Oduwole, E.O., Mahomed, H., Laurenzi, C.A., Larson, H.J., Wiysonge, C.S. 2021. Point-of-care vaccinators’ perceptions of vaccine hesitancy drivers: A qualitative study from the cape metropolitan district, South Africa, Vaccine39(39), pp. 5506-5512

Zhang, M.-X., Lin, X.-Q., Chen, Y., Tung, T.-H., Zhu, J.-S. 2021. Determinants of parental hesitancy to vaccinate their children against COVID-19 in China, Expert Review of Vaccines20(10), pp. 1339-1349

Coccia M. 2022. COVID-19 pandemic over 2020 (with lockdowns) and 2021 (with vaccinations): similar effects for seasonality and environmental factors. Environmental Research, Volume 208, 15 May 2022, n. 112711. https://doi.org/10.1016/j.envres.2022.112711

Verger, P., Peretti-Watel, P., 2021. Understanding the determinants of acceptance of COVID-19 vaccines: a challenge in a fast-moving situation. The Lancet. Public health 6 (4), e195–e196. https://doi.org/10.1016/S2468-2667(21)00029-3.

Ali, Z., Perera, S.M., Garbern, S.C., (...), Ali, J., Awada, N. 2022Variations in COVID-19 Vaccine Attitudes and Acceptance among Refugees and Lebanese Nationals Pre- and Post-Vaccine Rollout in Lebanon. Vaccines10(9),1533

Coccia M. 2021e. Pandemic Prevention: Lessons from COVID-19. Encyclopedia, vol. 1, n. 2, pp. 433-444. doi: 10.3390/encyclopedia1020036

Viswanath, K., Bekalu, M., Dhawan, D., et al., 2021. Individual and social determinants of COVID-19 vaccine uptake. BMC Publ. Health 21 (1), 818

Coccia M. 2022. Effects of strict containment policies on COVID-19 pandemic crisis: lessons to cope with next pandemic impacts. Environmental Science and Pollution Research, DOI: 10.1007/s11356-022-22024-w, https://doi.org/10.1007/s11356-022-22024-w

Sallam, M. 2021Covid-19 vaccine hesitancy worldwide: A concise systematic review of vaccine acceptance rates. Vaccines9(2),160, pp. 1-15

Coccia M. 2022. Improving preparedness for next pandemics: Max level of COVID-19 vaccinations without social impositions to design effective health policy and avoid flawed democracies. Environmental Research, vol. 213, October 2022, n. 113566. https://doi.org/10.1016/j.envres.2022.113566

Schwarzinger, M., Watson, V., Arwidson, P., Alla, F., Luchini, S., 2021. COVID-19 vaccine hesitancy in a representative working-age population in France: a survey experiment based on vaccine characteristics. Lancet Public Health published online Feb 5 https://doi.org/10.1016/S2468-2667(21)00012-8.

Machingaidze, S., Wiysonge, C.S. 2021. Understanding COVID-19 vaccine hesitancy, Nature Medicine 27(8), pp. 1338-1339

Author Response

We would like to thank the Reviewer 1 for his valuable comments, which made the manuscript a better version.

Abstract has to clarify empirical results and health and social implications of vaccination policy for the Lebanon.

Thank you. The Abstract was revised. Results were added. Also, a sentence about implication of vaccination policy was added.

Introduction has to better clarify the research questions of this study and provide more theoretical background. Authors have to better describe the role of vaccination to cope with COVID-19, optimal level  of vaccination and factors associated with vaccination. After that they can focus on the topics of this study to provide a correct analysis for fruitful discussion (See suggested readings that must be all read and used in the text). 

Thank you for the suggested readings. More background is added to the introduction.

“Lebanon is a small country in the Middle East. The healthcare system is mainly a private system. Lebanon's economy has been collapsing since 2019 and its currency has lost more than 90% of its value. There is no published data about adult vaccination in the country. However, after the Covid-19 pandemic, data was collected. The latest data of February 17, 2023 reveals the vaccination rate for 3 doses of Covid-19 to be 27.4%.”

Methods of this study is not clear. The section of Materials and methods must be re-structured with following three sections only and same order:

  • Sample and data
  • Measures of variables
  • Data analysis procedure. More details to provide.

Thank you. The Methods section is revised, and is now divided into 3 sections as suggested.

Results.

Authors have to discuss if the sample is or not representative of population.

Thank you, this was added in the Methods section: “Taking an acceptable risk of error of 3%, a 95% confidence level, and a total population of 5millions, the minimal sample size was calculated to be n=998 on Epiinfo software. We successfully managed to obtain the response of 1023 individuals which meets the sample size requirements of this study.”

Table 1 has also to indicate percentage of value. It would be better to represent these results with a bar graphs. 

Table 1 is changed to indicate numbers and percentages.

Title of tables has to be put above the table and not below. Title has to provide more information about analyses done and indicate location and period of analysis, always. This aspect for all tables. 

Titles are put above tables. Location and period of analysis was added to all tables.

Table 2 and 3. Significance of X2 p-value has to be indicated with *, such as ***=p-value <.001, **=p-value <0.01, *=p-value <0.05

Thank you, Tables were adjusted according to this suggestion.

Discussion. 

First, authors have to synthesize the main results in a simple table to be clear for readers and then show what this study adds compared to other studies.

Thank you. The results for the knowledge questions and attitude questions were summarized in Tables 2 and 4 and added to the Results.

Also, a paragraph was added in the discussion to summarize the results.

Section of Limitations and conclusion can be merged. 

This was changed as requested.

Conclusion has not to be a summary, but authors have to focus on manifold limitations of this study and provide suggestions of health, crisis management  and social policy, as well as how nations can prevent and manage with a good governance and appropriate communication next pandemics with effective vaccination plans and nonpharmaceutical measures of control. 

Thank you. The conclusion is revised according to the above recommendation.

I strongly suggest improving the paper,  by using all comments (suggested papers included to read and use) that I will in-depth verify,  and maybe it can be considered. 

Thank you for the suggested papers. Additional information was added in the manuscript, the readings were added within the references.

Reviewer 2 Report

The study has merit, and the findings are interesting. However, the paper itself needs work. 

First, the introduction gives the reader no context about the health care system, rate of vaccination, or rates of vaccine preventable diseases in Lebanon. For readers outside of Lebanon, this is essential. 

The tables showing the chi-square p-values are unclear. They should follow more standard formats for presenting survey results with the numbers, percents shown for each survey item and then the chi-square. 

The results are difficult to follow, but again, I believe the project has merit and should be published after the paper is significantly reformatted. 

Author Response

We would like to thank the Reviewer 2 for his valuable comments, which made the manuscript a better version.

The study has merit, and the findings are interesting. However, the paper itself needs work. 

First, the introduction gives the reader no context about the health care system, rate of vaccination, or rates of vaccine preventable diseases in Lebanon. For readers outside of Lebanon, this is essential. 

Thank you. All these suggestions were added in the Introduction.

“Lebanon is a small country in the Middle East. The healthcare system is mainly a private system. Lebanon's economy has been collapsing since 2019 and its currency has lost more than 90% of its value. There is no published data about adult vaccination in the country. However, after the Covid-19 pandemic, data was collected. The latest data of February 17, 2023 reveals the vaccination rate for 3 doses of Covid-19 to be 27.4% [13]. No studies about vaccination were done in Lebanon.”

The tables showing the chi-square p-values are unclear. They should follow more standard formats for presenting survey results with the numbers, percents shown for each survey item and then the chi-square. 

Tables were corrected as suggested. New tables were created to show number and percentages, and tables that show correlation were reviewed and changed as per your suggestions.

Reviewer 3 Report

Sakr and his colleagues conducted a survey to study the perception of adults about vaccination in Lebanese.They found most adults are concerned about the side effects of vaccines and their attitude towards vaccination were affected by their concerning.

Method-It is not clear how the samples were selected. Are there and including or excluding criteria?

Results-Tables 1, 2 and 3, it will be very helpful for the readers to get information if the authors could summarize their data in tables followed by p value calculation in each comparative groups.  

Limitations-Line 223-The authors addressed that this study was conducted in one center. How did they select this center? Is there bias when they select the samples since the majority of the participants were graduate 45.9% or post graduate level 24.8%, the majority of participants were healthcare professionals 27.6%, followed by business professions 18.1 %, architecture 11.2 %... (lines 99-101)

Author Response

We would like to thank the Reviewer 3 for his valuable comments, which made the manuscript a better version.

Method-It is not clear how the samples were selected. Are there and including or excluding criteria?

Thank you, Including and excluding criteria were added in The Method Section.

“Adults visitors and patients presenting to the clinics or the Emergency of the Lebanese American University Medical Center-Rizk Hospital (LAUMCRH) between January 2020 and January 2021 were asked if they are willing to participate and if agreed, were included in the study. Family medicine and Emergency Residents were approaching the participants and assisting them in filling the questionnaire.”

Results-Tables 1, 2 and 3, it will be very helpful for the readers to get information if the authors could summarize their data in tables followed by p value calculation in each comparative groups.  

Thank you for the suggestions. All tables were revised as per the reviewers suggestions.

Table 1 was changed to include number and percentages.

Tables 2 and Table 3 are now Table 3 and Table 5, they were changed according to reviewer 1 suggestion, X2 values were added, with p value marked by * when significant.

New Tables 2 and 4 were added.

Limitations-Line 223-The authors addressed that this study was conducted in one center. How did they select this center? Is there bias when they select the samples since “the majority of the participants were graduate 45.9% or post graduate level 24.8%, the majority of participants were healthcare professionals 27.6%, followed by business professions 18.1 %, architecture 11.2 %...” (lines 99-101)

It is true that the study was conducted in this center only. This is one limitation that we mentioned. First, subjects were recruited randomly while presenting to the clinics or to the Emergency and during the long period of time.

And second, the sample size was calculated to be representative of the population. The minimal sample size was calculated to be n=998 on Epiinfo software. We successfully managed to obtain the response of 1023 individuals which meets the sample size requirements of this study.

Reviewer 4 Report

In this manuscript Dr Rania Sakr and coauthors explored the perception of the Lebanese adults towards vaccination, and their knowledge and awareness towards its importance. The authors found that yhe majority of Lebanese individuals lack some knowledge about adult vaccination protection and benefits. The paper is interesting. However, I have some concerns.

Major

1) the survey was administered between 2020 and 2021 when Covid-19 vaccination had not yet began. However, since Covid-19 vaccines have unluckily increased the hesitancy towards immunization practice authors should discuss this important and actual point 

2) One of the most important point of the core of the no-vax galaxy is that not only vaccination is dangerous to health but also it does,t work and protect. In this connection, please provide data regarding the rate of immunization after vaccination in general, and Covid-19 in the specific: To this aim, please include and comment the following report:  doi:  10.3390/vaccines10020141. 

3) Table 2: Knowledge of subjects based on Chi2 p value; this table is rather obscure. Chi square analysis is related to a 2x2 tables. In this table it is not clear how many domains have the variables. For example, in which manner is explored the relathionship between aging and the opinions on Vaccines?

4) An important limitation that must be included in this manuscript is the paucity of metabolic and anthropometric covariates respect to the thousand of participants. It is well known in fact that obesity, diabetes and in general Cv risk factors may influence both vaccination efficacy (doi: 10.3389/fendo.2022.898810) and the severity of several infections (doi: 10.1007/s40618-020-01397-0)

Author Response

We would like to thank the Reviewer 4 for his valuable comments, which made the manuscript a better version.

1) the survey was administered between 2020 and 2021 when Covid-19 vaccination had not yet began. However, since Covid-19 vaccines have unluckily increased the hesitancy towards immunization practice authors should discuss this important and actual point 

It is true that the study was conducted before the Covid vaccine. A sentence about this was added in the discussion: ” These results would have certainly changed if the study was conducted years later, after the Covid-19 vaccine, knowing that the Covid-19 vaccines have unluckily increased the hesitancy towards immunization practice.”

2) One of the most important point of the core of the no-vax galaxy is that not only vaccination is dangerous to health but also it does,t work and protect. In this connection, please provide data regarding the rate of immunization after vaccination in general, and Covid-19 in the specific: To this aim, please include and comment the following report:  doi:  10.3390/vaccines10020141. 

Thank you. A paragraph was added “Another concern for the population is the protection offered by vaccines as well as the need for booster doses. Even after knowing that vaccination provide high level of protection, people have fear of the vaccines and are questioning their efficacy especially when we deal with new vaccines, as with the Covid-19 vaccination [16].”

 3) Table 2: Knowledge of subjects based on Chi2 p value; this table is rather obscure. Chi square analysis is related to a 2x2 tables. In this table it is not clear how many domains have the variables. For example, in which manner is explored the relathionship between aging and the opinions on Vaccines?

Tables were adjusted, and new tables were added.

4) An important limitation that must be included in this manuscript is the paucity of metabolic and anthropometric covariates respect to the thousand of participants. It is well known in fact that obesity, diabetes and in general Cv risk factors may influence both vaccination efficacy (doi: 10.3389/fendo.2022.898810) and the severity of several infections (doi: 10.1007/s40618-020-01397-0)

Thank you. A paragraph was added in the Limitation section about this: “Also, an important limitation is the paucity of metabolic covariates as example obesity, diabetes and other cardiovascular risk factors that may influence vaccination efficacy [18].”

Reviewer 5 Report

The manuscript by Sakr et al presents an important issue, particularly in nowadays with the new corona vaccine and the quite high levels of opponents.

Minor comments:

1- In the abstract please add the number of participants and the years of study.

2- English editing is required

3- Why there is no mention for the corona vaccine? 

Author Response

We would like to thank the Reviewer 5 for his valuable comments, which made the manuscript a better version.

In the abstract, please add the number of participants and the years of study.

Thank you for this suggestion, this was added in the abstract.

Why there is no mention of the corona vaccine.

The study was conducted before the Covid vaccine. However, the authors added new references about the Covid-19 vaccine.

Round 2

Reviewer 1 Report

I have read thoroughly the revised version of paper.

However, not all suggested papers have been read and used, such that theoretical framework is still week.

Read and use all suggested papers in my first report to improve the theoretical framework and discussion, 

After that the paper can be considered. 

Thanks

Author Response

The authors would like to thank Reviewer 1 for his valuable comments.

Thank you for the suggested references. They were all read and added to the manuscript. 

Reviewer 4 Report

No more requests

Author Response

The authors would like to thank Reviewer 4 for his valuable comments